

# Using a Virtual Machine environment for developing, testing, and training for the UM-UKCA Composition-Climate Model, using Unified Model version 10.9 and above

Nathan Luke Abraham[1,2], Alexander T. Archibald[1,2], Paul Cresswell[3], Sam Cusworth[3], Mohit Dalvi[3], David Matthews[3], Steven Wardle[3], and Stuart Whitehouse[3]

1. Department of Chemistry, University of Cambridge, Cambridge, CB2 1EW, UK
2. National Centre for Atmospheric Science, UK
3. Met Office, FitzRoy Road, Exeter, EX1 3PB, UK

*Correspondence to*: N. Luke Abraham (luke.abraham@atm.ch.cam.ac.uk)

**Abstract.** The Met Office Unified Model (UM) is a state-of-the-art weather and climate model that is used operationally worldwide. UKCA is the chemistry and aerosol sub model of the UM that enables interactive composition and physical atmosphere interactions, but which adds an additional 120,000 lines of code to the model. Ensuring that the UM code and UM-UKCA (the UM running with interactive chemistry and aerosols) is well tested is therefore essential. While a comprehensive test harness is in place at the Met Office and partner sites to aid in development, this is not available to many UM users. Recently the Met Office have made available a Virtual Machine environment that can be used to run the UM on a desktop or laptop PC. Here we describe the development of a UM-UKCA configuration that is able to run within this Virtual Machine while only needing 6GB of memory, before discussing the applications of this system for model development, testing, and training.

## 1 Introduction

The Met Office Unified Model (UM; Cullen, 1993) is a state of the art general circulation model, that is used operationally worldwide for weather forecasts and climate projections. The UM comprises of over 965,000 lines of computer code, mainly written in Fortran 90/95, and a comprehensive test harness is in place at the Met Office and some partner sites such as Australia, New Zealand, South Korea, Poland, and elsewhere, to aid in development and to ensure that the code is well-tested, which is comprehensive but labour intensive to maintain. There are over 450 UM developers, and this testing system is not available to many of those that do not work at the Met Office. There are three UM releases per year, and code must be tested as part of the standard code development process. The UM code itself is hosted at the Met Office Science Repository Service (U.K. Met Office, 2018c), which is accessible to all users.

Recently the Met Office have provided a Virtual Machine (VM) configuration that can be used to run the UM on a desktop or laptop PC (Matthews, 2018). A VM is essentially an emulation of a complete and self-contained computer (or *guest*), running as software within another computer (or *host*) which virtualises the hardware that the guest operating system is installed on.



A similar system is a *container*, which will only include the software stack required to execute a program that is then portable between different computers without needing a complex compile and install process. These virtual environments are useful when using large and complex computer models as they are standardised and consistent between different host platforms, although changes to the underlying hardware means that results may not be identical between simulations on two different

hosts using otherwise identical guest machines. As the environments are consistent and repeatable, these are often useful for training.

The Weather Research and Forecasting model (WRF; Skamarock et al., 2008) has been run on both VMs and containers (Hacker et al., 2016). Due to the standardised environment that they provide, VMs are also useful to enable analysis of model data by providing a consistent platform for analysis tools that is available to all researchers, such as that provided by the

Regional Climate Model Evaluation System (RCMES), which has also been used for training users of this system (NASA Jet Propulsion Laboratory, 2018). Here we describe the implementation of the United Kingdom Chemistry and Aerosols (UKCA) composition-climate model within the Met Office VM to better facilitate development and testing of new model code, and to provide a standardised environment for training new users in how to use UM-UKCA.

## 2    The Met Office Virtual Machine

The Met Office Virtual Machine makes use of VirtualBox (Oracle, 2018) and Vagrant (HashiCorp, 2018) to provision the VM and will automatically install the required packages that are needed to use the UM, namely FCM (U.K. Met Office, 2018a), Rose (U.K. Met Office, 2018d), and Cylc (NIWA, 2018). FCM is a build and version control system built around Subversion. Rose is a collection of tools that aim to improve the use of scientific computer models, both for research and operational use, and it is used as the graphical user interface for the UM to configure the input namelists. Cylc is a workflow tool that is used

to schedule the various tasks needed to run an instance of the UM in the correct sequence. With these installed it is then possible to install the UM on the VM, and this process is documented in detail in UM documentation paper X10 (Cresswell, 2018). It is also possible to install the Met Office Iris Python library (U.K. Met Office, 2018b) on the VM, which is used to read and process UM output files. The time taken to install everything and get the UM up and running is dependent on host machine and internet connection speed, but can be as quick as 30 minutes.

Some simple UM configurations are provided that only require 2 cores and 3GB of memory in total, based around low resolution versions of the Global Atmosphere 6 (Walters et al., 2017a), limited area, or the single-column model configuration of the UM (Cullen, 1993). However, configurations using UKCA chemistry and aerosol schemes usually require much more memory due to the large number of three-dimensional fields used.

For this study we use an Ubuntu 16.04LTS host with 32 Intel Xeon 2.0GHz cores and 64GB of memory, and an Ubuntu

16.04LTS guest configured with 2, 4, 8, or 16 cores and up to 20GB of memory as required. The Met Office VM can run a number of different guest GNU/Linux operating systems, although Ubuntu 16.04LTS is the current recommended distribution. It can also be used with GNU/Linux, macOS, and Windows hosts and will take up a maximum of 30GB of hard-disk space.



There is only a single user on the VM (named *vagrant*), and for simplicity the UM is installed into this user's home directory, rather than to a central location on the VM's filesystem (as would need to be done on a HPC system). Due to license restrictions, only freely available Fortran and C compilers can be used on the VM. The GNU gfortran and gcc compilers are used as standard as they come with the guest Ubuntu distribution used by this system and have adequate feature support to

compile the UM.

## 3   The UM-UKCA model

The United Kingdom Chemistry and Aerosols model is part of the Met Office Unified Model and provides a framework for adding aerosol and chemistry schemes to the UM. Aerosol microphysics is simulated by the GLOMAP-mode model (Mann et al., 2010), and this can be run with prescribed oxidant fields as in Global Atmosphere 7 (GA7; Walters et al., 2017b), or

coupled interactively to the different UKCA chemistry schemes provided. The chemistry scheme used here is *StratTrop* (also known as *CheST*), which is a combination of the stratospheric chemistry scheme of Morgenstern et al., 2009 and the tropospheric chemistry scheme of O'Connor et al., 2014. This scheme has previously been used in standalone studies (Banerjee et al., 2014; Esenturk et al., 2018; Ferracci et al., 2018; Finney et al., 2016; Hardiman et al., 2017) and for the Chemistry Climate Model Initiative (Morgenstern et al., 2017), and will be used in the upcoming UK Earth system model, UKESM1. A

full technical description of UKCA can be found in UM documentation paper 84 (Abraham et al., 2018). UKCA makes up over 12.5% of the UM code, at just over 121,000 lines.

Here we will use the StratTrop chemistry added to a GA7 configuration, which will require an additional 169 transported chemical and aerosol tracers and other 3D fields over the GA6 configuration currently provided on the VM, along with extra diagnostics calculated at run-time. Typical climate configurations of UKCA use the so-called N96L85 model resolution,

1.875°×1.25° with 85 vertical levels up to 85km. This configuration requires around 120GB of memory, and so running on a standard desktop is not feasible. However, it is possible to use a lower resolution, and turn-off high-memory sections of code, to allow a model configuration to fit within the specifications of a standard desktop, as described in Section 4.

## 4   Developing a low memory UKCA configuration

### 4.1   Changes to the model configuration

Figure 1 shows the development of a low-memory UKCA configuration that is suitable to be used on the VM, with further details provided in Table 1. Taking a N96L85 climate configuration based around the Global Atmosphere 7 configuration of the UM with UKCA StratTrop chemistry included, and re-gridding this to N48L70 (3.75°×2.5° with 70 vertical levels up to 80km) allowed the required memory to be reduced to only 12GB. While this is a great improvement, it is still too much to allow the set-up to be used easily in a VM on a personal computer. The number of vertical levels can be reduced further to 38

(with a model top at 40km), as is used in the HadGEM2-ES configuration of the UM (Collins et al., 2011; O'Connor et al.,




2014), and the dynamical timestep increased from 20 to 30 minutes, which allows the model to only need 7.5GB. The memory requirements can be lowered even more by turning off unnecessary diagnostics and diagnostic sections to reduce the number of large three-dimensions arrays that need to be allocated, leaving a model configuration that only needs 4GB of memory. The other sharp peaks that are consistent between configurations are from sections such as dynamics, convection, or unavoidable

5   calls to the diagnostics routines, and so cannot be turned off to reduce the memory requirements further.

When combined with the VM's operating system, using this 4GB configuration means that it is possible to run UM-UKCA with only 6GB required in total. This total memory requirement increases to 8GB if stricter compiler checks are used (see Section 5). Figure 2 compares UKCA ozone from a N96L85 climate configuration and from the low-memory N48L38 configuration that can be used on the VM.

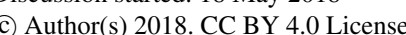

**Figure 1: Memory usage of various UM configurations on the VM. Each simulation was performed using 1 MPI process without using OpenMP parallelisation. Further details of these different configurations are given in Table 1.**

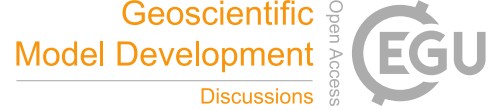



**Table 1: Description of the curves presented in Figure 1.**

| Curve in Figure 1 | UM resolution | Configuration details | Peak memory usage (GiB) 1×1 MPI processes without OpenMP |
|---|---|---|---|
| **Green curve** | N48L70 3.75°×2.5° with 70 vertical levels up to 80km | GA6 configuration with 27 dynamical timesteps (9 model hours). | 1.62 |
| **Black curve** | N48L70 3.75°×2.5° with 70 vertical levels up to 80km | GA7+StratTrop configuration with 3 dynamical timesteps and 1 chemical timestep (1 model hour). | 12.13 |
| **Purple curve** | N48L38 3.75°×2.5° with 38 vertical levels up to 40km | GA7+StratTrop configuration with 2 dynamical timesteps and 1 chemical timestep (1 model hour). | 7.43 |
| **Blue curve** | N48L38 3.75°×2.5° with 38 vertical levels up to 40km | GA7+StratTrop configuration with 2 dynamical timesteps and 1 chemical timestep (1 model hour). Only minimal diagnostic output is included. | 6.84 |
| **Red curve** | N48L38 3.75°×2.5° with 38 vertical levels up to 40km | GA7+StratTrop configuration with 2 dynamical timesteps and 1 chemical timestep (1 model hour). Only minimal diagnostic output is included and the CFMIP Observation Simulator Package (COSP) has been disabled. | 3.99 |





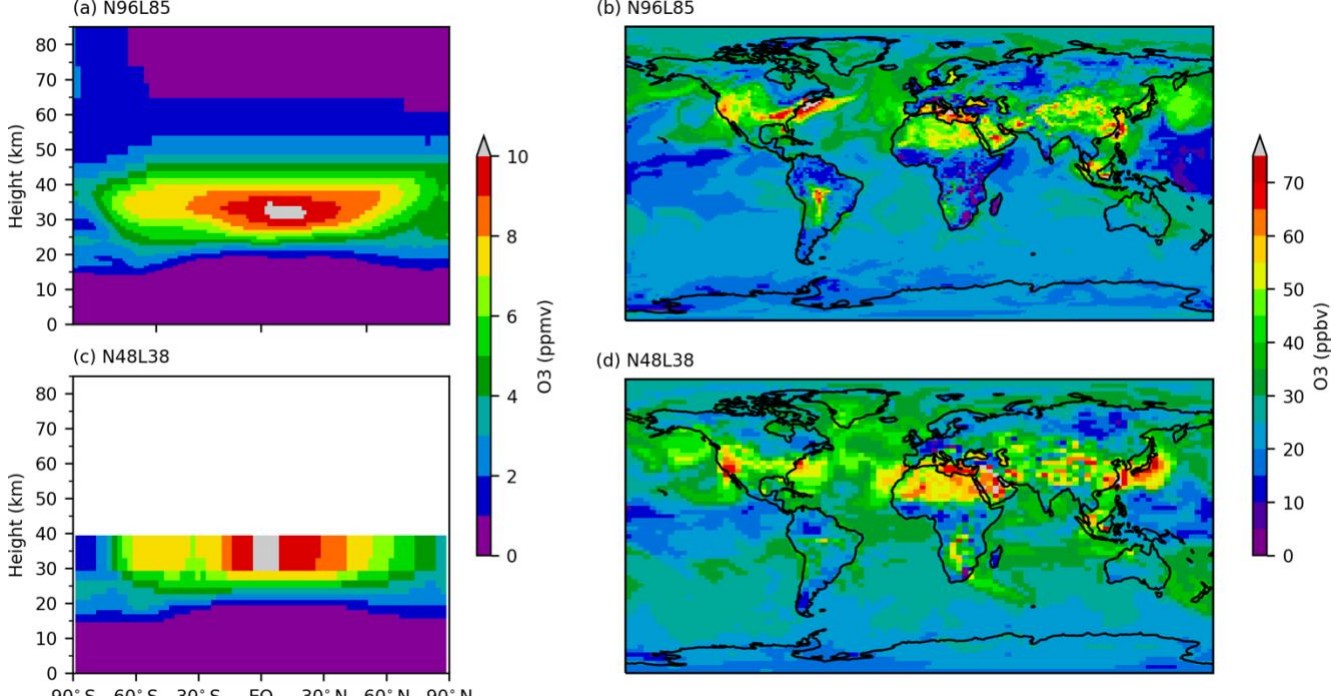

**Figure 2: (a, b) 3-hour mean ozone from N96L85 UM-UKCA configuration. (c, d) 3-hour mean ozone from N48L38 UM-UKCA configuration.**

## 4.2 Comparison to HPC implementation

Within the U.K., Academic researchers who use the UM have access to at least two supercomputers, ARCHER (a Cray XC30), and the Met Office XCS-C (a Cray XC40). Combined these have 57168 cores and 179136 GB of memory available for running the UM, and are both suitable for large production runs, such as for the CMIP6 historical and DECK experiments (Eyring et al., 2016), or for AerChemMIP (Collins et al., 2017).

However, not all work involves long and large simulations. When developing a change often what is most important is the
speed of compilation and the over-all turn-around from "what changes to I need to implement?" to "the model runs successfully".

Table 2 shows very approximate timings on these three different environments (ARCHER, the XCS-C, and the VM) for the compile (including code extract), reconfiguration, and atmosphere tasks for the N48L38 UKCA configuration described in Section 4. Care should be taken with these results, as they are dependent on many factors that are impossible to isolate. The
compile, reconfiguration, and atmosphere tasks may each use a different number of MPI processes on the three platforms, with different clock speeds and using different filesystems. Different compilers, compiler versions, and flags are used between the machines. Here the run-length has been increased to 3 model hours (6 dynamical timesteps, 3 chemical timesteps).





While the model run-time is fastest on the two supercomputers, the compile time is comparable on the VM. When queue times are considered the VM will be a better option for model development jobs over ARCHER, especially as more cores can be provided to the VM if available, reducing run-times (see Table 4).

Throughput on the XCS-C is broadly better than on ARCHER and the VM, although both of these supercomputers are large

and complex, and will always have delays due to planned (and unplanned) downtime for example. As the VM is a *single-user* environment that can be installed on a personal computer, the whole system is simpler and less error prone as there is no waiting, no queues, no filesystem I/O contention etc. This means that, for initial developments at least, the VM environment is likely to be the easiest and fastest to use due to the immediate turnaround.

## 5   Testing new model developments

Each working copy of the UM includes a complete Rose suite that can be used to test the UM or install it to a fixed location. This suite is used with the *rose-stem* utility, which is used to run such test suites; the test suite itself is known as rose-stem (Whitehouse, 2018) and contains a number of different applications, known as *apps*, which perform different tasks that can be automated. One of the functions of this suite is to install the UM itself, but it can also be used to run a series of known UM configurations and test them. As well as confirming that the tasks complete successfully, rose-stem is also able to compare

the output from these configurations to already created standard output (also called *Known Good Output*, or KGO). This forms part of the standard UM development process at the Met Office and several partner sites, as each site has its own rose-stem configuration, including the Virtual Machine. When run with a new development, rose-stem will enable a user to test that their code compiles and runs, and also whether it changes the results of known tests. Rose-stem is usually called from within a copy of the UM code; for installation this will be a copy of the UM trunk at a particular version (e.g. vn10.9), but for testing

developments this will need to be a local copy containing the desired changes. It should be noted that if model output is no longer identical to the current KGO, this does not mean that the new output is not scientifically valid, just that the changes made to the code have also changed the results. Further testing and validation will be required to determine the scientific impact of these code changes.




**Table 2: Rough timings for running UM version 10.9 N48L38 UKCA suite on ARCHER, the XCS-C, and the VM without using OpenMP. The UM could not be used with GNU gfortran on ARCHER as some dependencies as not been made available with this compiler.**

| Step | ARCHER (XC30) | XCS-C (XC40) | Virtual Machine |
|---|---|---|---|
| Cray `cce` initial compile | 34 minutes | 15 minutes | - |
| Cray `cce` incremental compile | 5-7 minutes | 3 minutes | - |
| Intel `ifort` initial compile | 19 minutes | 9 minutes | - |
| Intel `ifort` incremental compile | 6-7 minutes | 1 minute | - |
| GNU `gfortran` initial compile | - | 4 minutes | 8-10 minutes |
| GNU `gfortran` incremental compile | - | 2-3 minutes | 45 seconds |
| Reconfiguration task, used to produce the initial conditions file. | 3-4 minutes (Intel, 6×4 processes) | 15-30 seconds (GNU, 4×9 processes) | 25-30 seconds (GNU, 1×2 processes) |
| Atmosphere task | 40 seconds (Intel, 6×4 processes) | 45 seconds (GNU, 4×9 processes) | 12 minutes (GNU, 1×2 processes) |

At UM version 11.0 there are 109 different configurations of the UM tested at the Met Office with rose-stem, with more at

5    partner sites, which test different science, compiler, and machine settings. On the Virtual Machine there are only 12 UM tests, with 3 of these being UKCA configurations. On the VM the UKCA rose-stem jobs provide the following checks:

1. Test that new code compiles without errors when using both *safe* and *rigorous* compiler settings (see Table 3).
2. Tests that new code runs without errors when using both *safe* and *rigorous* compiler settings (see Table 3).

10   3. Tests whether new code changes results (this may be expected, depending on the change made).
4. Tests whether new code changes maintain results when using different processor decompositions.
5. Tests whether OpenMP parallelisation has been affected by new code changes, which then change results.



**Table 3: UKCA rose-stem jobs available on the VM, showing the differences between Fortran compiler flags used for safe and rigorous compile options. C flags also differ. Both safe jobs point to the same KGO files.**

| Rose-stem job name within the `ukca` group | Compile type | GNU `gfortran` compiler flags | Number of UM OpenMP threads | Total VM memory required (GB) |
|---|---|---|---|---|
| `vm_n48_ukca_eg_noomp` | safe | `-O2 -Werror` | 0 | 6 |
| `vm_n48_ukca_eg_omp_noios` | safe | `-O2 -Werror` `-fopenmp` | 2 | 6 |
| `vm_n48_ukca_eg_omp_noios_comp_check` | rigorous | `-O0 -Wall` `-ffpe-trap=invalid,zero` `-fbounds-check` `-Warray-bounds` `-fcheck-array-temporaries` `-finit-real=nan` `-fimplicit-none` `-fopenmp` | 2 | 8 |

A listing of the UKCA rose-stem jobs on the VM can be found in Table 3. Each of these is constructed using a *graph*, which

5   links the different tasks of the job together by their dependencies. A simplified graph of these UKCA rose-stem tests at UM version 11.0 is shown in Figure 3. Additionally, equivalent GNU N48L38 tests to those on the VM have been implemented at the Met Office and are performed daily against the UM trunk. These test if any new code changes are likely to break the tests on the VM, effectively reducing the need for frequent VM rose-stem testing against the current and most recent UM code.



**Table 4: Approximate timings for different UKCA configurations on the VM at UM version 11.0. The compiler settings are listed in detail in Table 3.**

| Compiler settings | Safe | Safe | Rigorous |
|---|---|---|---|
| **Number of UM OpenMP threads** | **0** | **2** | **2** |
| **Approximate run-time on 2-core VM (1×2) (minutes)** | 8 | 11 | 29 |
| **Approximate run-time on 4-core VM (1×2) (minutes)** | 8 | 4 | 17 |
| **Approximate run-time on 4-core VM (1×4) (minutes)** | 5 | 8 | 22 |
| **Approximate run-time on 8-core VM (1×4) (minutes)** | 5 | 3 | 14 |
| **Approximate run-time on 8-core VM (1×8) (minutes)** | 3 | 6 | 26 |
| **Approximate run-time on 16-core VM (1×8) (minutes)** | 3 | 3 | 22 |

## 5.1 Known Good Output tests

In order to test against already produced *Known Good Output* (KGO), this KGO must be produced and installed into a standard

location. At the Met Office, this step is performed manually when new code is committed to the UM trunk, and so a growing and continual series of KGO is maintained and available; this allows any unexpected change of results to be detected as changes are made to the trunk and facilitates the testing of changes to the UM at or near the latest trunk revision. However, for the VM this maintenance, while possible, is labour intensive and usually unnecessary. The VM is not designed as a system that should be maintained through UM versions, it is designed to be used for simple code development against a single, stable UM release.

This is also important due to other possible changes to the UM, Rose, and Cylc systems that would also be time-consuming to maintain. On the VM the KGO for each rose-stem job must be generated and installed when the UM is installed as it is machine dependent. This can be done by using the command

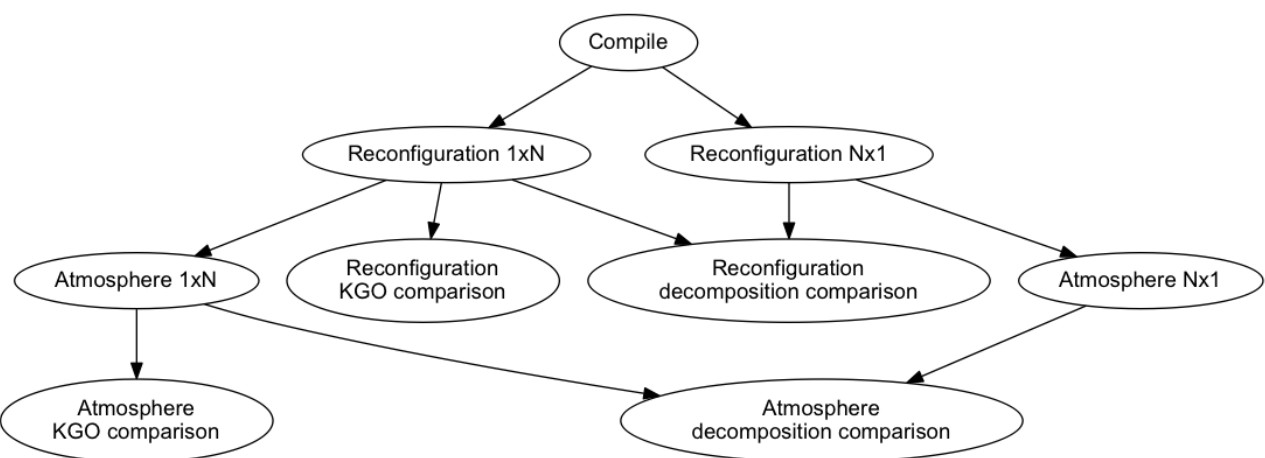

**Figure 3: Simplified example rose-stem graph, showing the different tasks performed during a single job.**

```
rose stem --group=ukca -S GENERATE_KGO=true
```

when running rose-stem on a copy of the UM trunk at the required version. The `ukca` term refers to the group of all the UKCA rose-stem jobs available on the VM (listed in Table 3).

Once the KGO for the UKCA jobs is in place, it is then possible to run the rose-stem tests on a local copy of the UM containing a code change by using the command

```
rose stem --group=ukca
```

within the top-level directory of the local copy of the UM code. It is also possible to run separate jobs individually by replacing the `ukca` above by the rose-stem job name (see Table 3). Once the various tasks are complete, rose-stem will automatically perform a bit-wise comparison between the UM output files generated by the reconfiguration and UM atmosphere steps and will highlight any differences between the KGO files and the files produced from the rose-stem jobs. Often major science

changes or bugfixes will result in differences, but small changes or new schemes that are turned off should usually preserve the model evolution and hence the KGO tests should pass. If these KGO tests do not pass when they would be expected to do so, this may point to an error in the implementation.

## 5.2 Processor decomposition tests

From UM version 11.0, when the flag `-S INTEGRATION_TESTING=true` is passed to rose-stem, the UKCA jobs can be

run in two different configurations to enable processor decomposition testing. Here the UM will use either 1×N processes or N×1 processes. By default, N=2, but this can be changed by adding `-S MPI_TASKS=`$N$ on the call to rose-stem. The output from these two jobs is then compared using mule-cumf in the same way as for the KGO comparison. By increasing the value of N it is possible to reduce the run-time of these configurations, as seen in Table 4.

Scientifically, maintaining results when changing the number (or decomposition) of is desirable as it allows for the resources

to be changed during long climate runs, allowing for more or less MPI processes to be used without affecting the model output. Technically, it helps test for code correctness. There may, for example, be bounds errors or race conditions that would never be encountered if only one decomposition is routinely tested, but that may show up if the run conditions are varied. By ensuring several decompositions give the same answer we reduce the risk of bugs in the code; something that changes answers in a second decomposition may cause a crash in third or produce nonsense.

Additionally, any impacts on OpenMP parallelisation can be tested by running job `vm_n48_ukca_eg_omp_noios`, as the KGO is produced using job `vm_n48_ukca_eg_noomp` and should be identical for both.



## 6   Training new users

When training new users of UM-UKCA how to use the model, we begin with the premise *"what are the most common things that a new PhD student or researcher would need to know how to do in order to use UKCA?"*. When starting with UM-UKCA, often new users will wish to perform studies that will involve a combination of the following:

- Creating or modifying emissions
- Adding new chemical species
- Adding new reactions
- Adding new deposition processes
- Adding new diagnostics

as well as needing to learn how to output and process UM and UKCA diagnostics, and also wanting to become more familiar with the model's user-interface.

We can in fact cover all these points by imagining two new chemical species, ALICE and BOB, and the chemical equation

$$ALICE + OH \rightarrow BOB + Secondary\ Organic\ Species \tag{R1}$$

where *Secondary Organic Species* will condense to form secondary organic aerosol (Mann et al., 2010), ALICE is dry deposited, and BOB is wet deposited. The specifics of the emission field, reaction and deposition rates are unimportant, so

long as they have reasonable values.

The steps to include the toy chemistry above to UM-UKCA can then be broken down into the following tasks:

1. Add ALICE and BOB as transported tracers to the UM and UKCA
2. Add surface emissions of ALICE, comprising:
a. Re-gridding of a provided 1°×1° dataset to N48 resolution
    b. Saving this re-gridded emissions field to NetCDF with the correct metadata required by UKCA
    c. Including this new NetCDF emissions file in the UKCA namelist within Rose
3. Add the bimolecular reaction R1 to the UKCA StratTrop chemical mechanism
4. Add the dry-deposition of ALICE to UKCA
5. Add the wet-deposition of BOB to UKCA
6. Add new diagnostic reaction fluxes to UM-UKCA to output the fluxes through steps introduced in 3, 4, and 5 above
7. Process UM output to calculate aerosol optical depth



To aid new users of UM-UKCA a series of online tutorials have been developed that go through the various (and complex) steps above required to make changes to the model. Detailed instructions are available for UM versions 8.2 (Abraham, 2013), 8.4 (Abraham and Mann, 2014), 10.4 (Abraham and Mann, 2016), and 10.9 (Abraham et al., 2017) using ARCHER. Additionally, at UM version 10.9 this UKCA training can be performed exclusively on the VM using a configuration based on the N48L38 model described in Section 4. Figure 4 shows typical output from the above tasks after they have been completed.







**Figure 4: Showing output from the various UKCA tutorial tasks. (a) 1°×1° supplied September emissions. (b) Emissions re-gridded to N48. (c) 3-hour mean ALICE surface mole fraction after the tutorials have been completed. (d) as for (c), but for BOB. (e) 3-hour mean column-integrated flux through reaction R1. (f) 3-hour mean column-integrated ALICE dry deposition flux. (g) 3-hour mean column-integrated BOB dry deposition flux. (h) 3-hour mean total Aerosol Optical Depth from GLOMAP-mode at 0.55μm. ALICE and BOB were both initialised to $1.0 \times 10^{-12}$ kg kg$^{-1}$.**



## 7 Conclusions

As can be seen in Table 2, the UM-UKCA configurations on the VM require longer runtimes than equivalent configurations on HPC systems. Using 2 MPI processes the N48L38 configuration would take at least 32 hours to simulate one model month, although using a server as a host and adding more processes reduces this as seen in Table 4.

The decrease in horizontal resolution will be somewhat detrimental to the representation of the model dynamics and other physical or chemical processes (Stock et al., 2014; Strachan et al., 2013), although many UKCA studies have been performed using N48 resolution (Banerjee et al., 2014; Bednarz et al., 2016; Keeble et al., 2017; Nowack et al., 2014). However, the reduction in the number of vertical levels (see Figure 2) is a significant issue as the chemistry scheme used, StratTrop, is designed to simulate both stratospheric and tropospheric chemistry, and without a full model stratosphere it is computationally

inefficient to include all the reactions in this scheme. Also, due to this low top, the look-up table photolysis scheme, used in the region above the Fast-JX photolysis scheme (Telford et al., 2013), will not be called.

The chemistry scheme used in HadGEM2-ES (O'Connor et al., 2014) would be appropriate for this level structure, but this would then not test the equivalent settings to the N96L85 GA7+StratTrop configuration, and so the usefulness of the tests listed in Table 4 would be reduced. By keeping as many settings as possible the same between these configurations it allows

the VM tests to be traceable to N96L85 tests performed on the Met Office systems.

While the VM is ideal for initial development and testing of new code, once long science integrations need to be done it is necessary to switch to a standard N96L85 climate configuration on a HPC system. Simulations of around 20-years in length are usually required to see if any new science introduces a significant change to stratospheric composition, and these are not feasible to be performed on the VM. The VM could however be used for stand-alone scientific studies making use of the

UKCA Box Model (Esenturk et al., 2018) or UKCA called from the Single Column Model, although further work would be required to allow these configurations to work on the VM.

While the toy chemistry discussed in Section 6 does not cover all reaction types covered and may be considered simpler than most changes new users may wish to make, it is sufficiently detailed enough to give users experience with many different parts of the UKCA code. While using the VM will give users experience of using and editing the UM's Rose graphical user interface,

it will not provide experience in using UM-UKCA on a HPC system. However, this is also an advantage, as it means that it is possible to gain experience in using and developing UM-UKCA before committing supercomputer resources.

Despite these limitations, the VM is an easy-to-use system that is available to all current and potential users, and creates a consistent environment for model development, testing, and training. This system is complementary to, rather than being a replacement for, using UM-UKCA on HPC systems.

## 30 Code Availability

Due to intellectual property right restrictions, we cannot provide either the source code or documentation papers for the UM.





*Obtaining the UM.* The Met Office Unified Model is available for use under licence. The functionality discussed here is fully available in the UM trunk from version 11.0, with instructions on using the VM provided by UMDP X10 (Cresswell, 2018), and the UM version 10.9 UKCA Tutorials are described in Abraham et al., 2018a. A number of research organisations and national meteorological services use the UM in collaboration with the Met Office to undertake basic atmospheric process
research, produce forecasts, develop the UM code and build and evaluate Earth system models. For further information on how to apply for a licence see http://www.metoffice.gov.uk/research/modelling-systems/unified-model.

**Acknowledgements**

This work used the ARCHER UK National Supercomputing Service (http://www.archer.ac.uk). This work used Monsoon2, a collaborative High-Performance Computing facility funded by the Met Office and the Natural Environment Research
Council. This work used the NEXCS High Performance Computing facility funded by the Natural Environment Research Council and delivered by the Met Office.

UKCA training has been supported by the NERC ACITIES atmospheric chemistry modelling network, grant number NE/K001280/1 and NERC Advanced Training Short Courses scheme, grant numbers NE/M006220/1, NE/N000129/1, NE/N019091/1, and NE/P020089/1. We would like to thank the UM Systems Team at the Met Office. NLA would like to
thank John Pyle, Bryan Lawrence, and NCAS Computational Modelling Services for supporting this work.

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
