# Peer review of "Using a Virtual Machine environment for developing, testing, and training for the UM-UKCA Composition-Climate Model, using Unified Model version 10.9 and above"

_Geoscientific Model Development, 2018_

## Referee Comment (RC1) · Anonymous Referee #1 · 14 Jun 2018

General Comments:

This manuscript describes a virtual machine (VM) for low-resolution testing and code development of the Met Office Unified Model, focusing on the chemistry and aerosol sub-module. The use of VM's and containers remains relatively new in the geophysical model development community, and the exploration presented here is a worthwhile contribution. The contribution is generally well-written and complete, subject to some clarifications requested below.

Specific Comments:

While the manuscript is generally well written, I found Section 4.2 somewhat confusing and possibly unnecessary. The comparison between two HPC platforms and a 1x2 decomposition in the VM did not lead to any conclusions that I could discern. I can't see how the statement that the "compile time is comparable on the VM" is supported by the evidence presented. I am also unable to find a description of where the VM is deployed in these tests.

The relevant issue here is the overhead of the VM, which we can expect will be large compared to e.g. a container. A 1x2 HPC test, then a 1x2 test with the VM deployed on the same HPC, would directly address that question.

A second comment is that while the results are likely reproducible with a few limitation of hardware (maybe), the general lack of access to the code due to restricted access does diminish the value of this contribution.

Technical corrections:

P1L22: Break up the sentence beginning with "The UM..."

P2L15: Please clarify the discussion of Rose and Cylc. As written they both appear to be workflow management systems of some kind.

P4L1: How does the time step affect memory? I can't follow that sentence.

P10L8: I suggest a new paragraph at "The VM"

P11L22: "mule-cumf" needs explanation

---

## Referee Comment (RC2) · Anonymous Referee #2 · 14 Jun 2018

Abraham et al. (2018) describe how a global atmospheric chemistry-climate model (UM-UKCA) was modified for use in a virtual machine with a low memory footprint, making it easily portable to most modern desktop or laptop computers with the appropriate host software installed. The resolution of the model was significantly reduced, and a number of other features were disabled, rendering the model configuration unsuitable for production use. The point of the exercise however was to provide an environment suitable for certain kinds of model development and testing, and for training new users of the system.

The modifications to the base model are described, and the different use cases for this system are discussed. This paper should be interesting and useful for users of the virtual machine environment, and other workers who are also considering packaging such a model in a virtual environment. I recommend publication in GMD after a few minor revisions.

In your "Code Availability" section, you describe the restrictions and availability of the UM, but you do not mention anything about obtaining a VM image, except for a cryptic reference to "UMDP X10", which is not described any further. It might seem a bit pedantic, but since the whole paper is about this virtual machine, it would be nice to see some more text about its availability.

Page 4, line 2: Which diagnostics were turned off, and why are these "unnecessary"?

Page 4, line 9: A short discussion of the differences in the ozone simulated by the two configurations would be appropriate here. To put the changes into context, perhaps these differences could be compared with the multi-model spread in any of the recent model intercomparison exercises.

Table 1: I found the "Configuration details" of the black and purple curves slightly confusing. Instead of giving the number of timesteps in one model hour, it seems more natural to give the length of the timestep (eg. 20 minutes and 30 minutes).

Figure 2: By plotting the results of the full and reduced configurations next to each other, the reader is required to eyeball the two figures to get a sense of the differences between the simulations. For panels (b) and (d), I would prefer to see the plot of the full configuration as-is, and then a plot of the difference between the reduced and full versions. Also, what does "3-hour mean" mean in this context? Is this a single 3-hour period from one particular day? More information is needed here.

Page 6, line 10: The first "to" should be "do".

Page 11, line 22: What is "mule-cumf"?

[Figure]

Page 11, line 24: "of is desirable". I think you left out some words here.

Page 13, line 1: Can you provide some kind of link to the online tutorials? Ideally in a permanently-available archive? Or could you zip them and add them as a supplement to the paper?

---

## Author Comment (AC1) · 20 Jul 2018

*Response to the Interactive Comments:*

**Using a Virtual Machine environment for developing, testing, and training for the UM-UKCA Composition-Climate Model, using Unified Model version 10.9 and above**

Nathan Luke Abraham[1,2], Alexander T. Archibald[1,2], Paul Cresswell[3], Sam Cusworth[3], Mohit Dalvi[3], David Matthews[3], Steven Wardle[3], and Stuart Whitehouse[3]

1. Department of Chemistry, University of Cambridge, Cambridge, CB2 1EW, UK
2. National Centre for Atmospheric Science, UK
3. Met Office, FitzRoy Road, Exeter, EX1 3PB, UK

*Correspondence to*: N. Luke Abraham (luke.abraham@atm.ch.cam.ac.uk)

We would like to thank both reviewers for taking the time to read the manuscript. We include their full comments below in ***bold italics***, with our responses interspaced below in normal text. Changes to the manuscript text are given in *italics*.

**Response to Anonymous Referee #1**

*General Comments:*

***This manuscript describes a virtual machine (VM) for low-resolution testing and code development of the Met Office Unified Model, focusing on the chemistry and aerosol sub-module. The use of VM's and containers remains relatively new in the geophysical model development community, and the exploration presented here is a worthwhile contribution. The contribution is generally well-written and complete, subject to some clarifications requested below.***

We would like to thank the reviewer for their positive helpful comments.

*Specific Comments:*

***While the manuscript is generally well written, I found Section 4.2 somewhat confusing and possibly unnecessary. The comparison between two HPC platforms and a 1x2 decomposition in the VM did not lead to any conclusions that I could discern. I can't see how the statement that the "compile time is comparable on the VM" is supported by the evidence presented. I am also unable to find a description of where the VM is deployed in these tests.***

*The relevant issue here is the overhead of the VM, which we can expect will be large compared to e.g. a container. A 1x2 HPC test, then a 1x2 test with the VM deployed on the same HPC, would directly address that question.*

5    In Section 2 we state that "For this study we use an Ubuntu 16.04LTS host with 32 Intel Xeon 2.0GHz cores and 64GB of memory, and an Ubuntu 16.04LTS guest configured with 2, 4, 8, or 16 cores and up to 20GB of memory as required." and this is the case for VM integrations throughout the manuscript. The text has been altered to highlight this:

*Table 2 shows very approximate timings on these three different environments (ARCHER, the XCS-C, and the VM set-up*
10    *described in Section 2) for the compile (including code extract), reconfiguration, and atmosphere tasks for the N48L38 UKCA configuration described in Section 4.*

In Section 4.2 we wish to highlight how a Virtual Machine environment can be useful for model development by comparing how a similar model set-up, that has designed to be run quickly, can be used on the different platforms available. For this we
15    used the optimum configuration on each of the different systems, rather than using a 1x2 configuration everywhere, as we wanted to highlight the differences during a typical development cycle. It is for these reasons that only very rough timings are given. As the Reviewer highlights there is additional overhead when using the VM compared to using a container, but we did not wish to perform a detailed timing comparison as the configuration presented here is designed for model development and testing, and not for production simulations where model performance would need to be optimised. The additional text has been
20    added to emphasize this point:

*Here we are not considering a detailed comparison of identical N48L38 UKCA configurations on different computing systems but are instead interested in the typical configuration that would be used on each system when making new code developments.*

25    On the VM, GNU compiles took around 8-10 minutes for initial builds, with subsequent incremental builds taking less than a minute. On NEXCS GNU compiles took around 4 minutes for an initial compile and 2-3 minutes for subsequent incremental compiles. Times for the Intel and Cray Compiler Environment compiles were longer than both of these. We have amended the text to reduce the emphasis on the compile time:

30    *While the model run-time is fastest on the two supercomputers, when queue times are considered the VM may be a better option for model development jobs over ARCHER. More cores can also be provided to the VM if available, further reducing run-times (see Table 4).*

We would also be unlikely to be able to deploy the VM on to either HPC environments due to security concerns.

*A second comment is that while the results are likely reproducible with a few limitation of hardware (maybe), the general lack of access to the code due to restricted access does diminish the value of this contribution.*

5   As the Known Good Output is internally generated within the VM system, any changes in model evolution between different VM instances due to the differences in the Host hardware are accounted for during the rose-stem tests. As this system is designed to be a self-contained platform for model development, testing, and training, the lack of reproducibility of the results between different Hosts should not be an issue for users.

The nature of the UM licence means that its use is restricted. However, for those with access we believe that this will be a
10   useful tool, and there has already been an uptake in this VM system by UKCA and UM users.

We also wish to highlight how useful small systems, such as the VM environment described here, are for users and developers, without the need to use HPC resources.

*Technical corrections:*
15   *P1L22: Break up the sentence beginning with "The UM: : :"*

The text now reads:

*The UM comprises of over 965,000 lines of computer code, mainly written in Fortran 90/95. A comprehensive test harness is*
20   *in place at the Met Office and some partner sites such as Australia, New Zealand, South Korea, Poland, and elsewhere, to aid in development and to ensure that the code is well-tested, which is comprehensive but labour intensive to maintain.*

*P2L15: Please clarify the discussion of Rose and Cylc. As written they both appear to be workflow management systems of some kind.*

25

Cylc is a workflow engine, used to schedule the various tasks required, whereas Rose is used to configure the UM and edit the input namelists. We have amended the text in the manuscript to clarify this:

*Rose is a system for creating, editing, and running application configurations and it is used as the graphical user interface for*
30   *the UM to configure the input namelists.*

*P4L1: How does the time step affect memory? I can't follow that sentence.*

It is the reduction in model levels that reduced the memory, not the change in the model timestep. The reduction in the number of levels then allows the timestep to be increased, which is then beneficial for model run time. The sentence has been changed to read:

5 *The number of vertical levels can be reduced further to 38 (with a model top at 40km), as is used in the HadGEM2-ES configuration of the UM (Collins et al., 2011; O'Connor et al., 2014), which allows the model to only need 7.5GB of memory and allows the dynamical timestep to be increased from 20 to 30 minutes, also reducing the overall run-time.*

***P10L8: I suggest a new paragraph at "The VM"***

This change has been made. The text now appears:

*However, for the VM this maintenance, while possible, is labour intensive and usually unnecessary.*
*The VM is not designed as a system that should be maintained through UM versions, it is designed to be used for simple code*
15 *development against a single, stable UM release.*

***P11L22: "mule-cumf" needs explanation***

The mule utility is included as part of the UM install and contains a number of useful tools for examining UM files. One of
20 these tools allows for the comparison of UM files, called "mule-cumf", which allows for a bit-wise comparison of all fields contained in UM output files to be performed. However, in the context of this discussion these details are unnecessary, and so the reference to mule-cumf has been removed. The text now reads:

*The output from these two jobs is then compared in the same way as for the KGO comparison.*

25 **Response to Anonymous Referee #2**

***Abraham et al. (2018) describe how a global atmospheric chemistry-climate model***
***(UM-UKCA) was modified for use in a virtual machine with a low memory footprint,***
***making it easily portable to most modern desktop or laptop computers with the appropriate***
***host software installed. The resolution of the model was significantly reduced,***
30 ***and a number of other features were disabled, rendering the model configuration unsuitable***
***for production use. The point of the exercise however was to provide an environment***
***suitable for certain kinds of model development and testing, and for training***

*new users of the system.*

*The modifications to the base model are described, and the different use cases for this system are discussed. This paper should be interesting and useful for users of the virtual machine environment, and other workers who are also considering packaging*

5 *such a model in a virtual environment. I recommend publication in GMD after a few minor revisions.*

We would like to thank the reviewer for their positive and helpful comments.

10 *In your "Code Availability" section, you describe the restrictions and availability of the UM, but you do not mention anything about obtaining a VM image, except for a cryptic reference to "UMDP X10", which is not described any further. It might seem a bit pedantic, but since the whole paper is about this virtual machine, it would be nice to see some more text about its availability.*

15

Under the *Code Availability* section, we have included a comment regarding the availability of the Virtual Machine configuration settings, which are freely available via GitHub. As mentioned in this section, all Unified Model Documentation Papers, including UMDP X10, are available under the UM licence. This paper is not intended to be a replacement for UMDP X10, and the citations for Cresswell (2018) and all other UMDPs include the URLs of these document on the Met Office

20 Science Repository Service (U.K. Met Office, 2018). We have therefore clarified the text in the *Code Availability* section to:

*The Met Office Virtual Machine can be obtained from https://github.com/metomi/metomi-vms (Matthews, 2018). Due to intellectual property right restrictions, we cannot provide either the source code or documentation papers for the UM.*

25 *Obtaining the UM. The Met Office Unified Model is available for use under licence. The functionality discussed here is fully available in the UM trunk from version 11.0, with detailed instructions on using how to install the UM on the VM provided by UMDP X10 (Cresswell, 2018), and the UM version 10.9 UKCA Tutorials are described in Abraham et al. (2017). A number of research organisations and national meteorological services use the UM in collaboration with the Met Office to undertake basic atmospheric process research, produce forecasts, develop the UM code and build and evaluate Earth system models.*

30 *For further information on how to apply for a licence see http://www.metoffice.gov.uk/research/modelling-systems/unified-model.*

The UMDP citations appear as:

Abraham, N. L., Archibald, A. T., Bellouin, N., Boucher, O., Braesicke, P., Bushell, A., Carslaw, K., Collins, B., Dalvi, M., Dennison, F., Emmerson, K., Folberth, G., Haywood, J., Hewitt, A., Johnson, C., Kipling, Z., Macintyre, H., Mann, G., Telford, P., Merikanto, J., Morgenstern, O., O'Connor, F., Ordonez, C., Osprey, S., Pringle, K., Pyle, J., Rae, J., Reddington, C., Savage, N., Sellar, A., Spracklen, D., Stier, P., West, R., Mulcahy, J., Woodward, S., Boutle, I. and Woodhouse, M. T.: UMDP 084: United Kingdom Chemistry and Aerosol (UKCA) Technical Description. [online] Available from: https://code.metoffice.gov.uk/doc/um/vn11.0/papers/umdp_084.pdf (Accessed 23 February 2018), 2018.

Cresswell, P.: Unified Model Documentation Paper X10: Unified Model Virtual Machine Guide. [online] Available from: https://code.metoffice.gov.uk/doc/um/vn11.0/papers/umdp_X10.pdf (Accessed 6 March 2018), 2018.

Whitehouse, S.: Unified Model Documentation Paper X09: The UM rose-stem Suite for External Users. [online] Available from: https://code.metoffice.gov.uk/doc/um/vn11.0/papers/umdp_X09.pdf (Accessed 16 February 2018), 2018.

***Page 4, line 2: Which diagnostics were turned off, and why are these "unnecessary"?***

The Unified Model has a comprehensive diagnostics package and can output a large number of diagnostic fields from many different UM code sections. Many UM climate configurations contain a standard set of diagnostics that are included in long 20-year simulations that are then processed to produce a series of standard plots to aid in model evaluation. The original N48L70 GA7+StratTrop configuration contained 695 separate diagnostic requests, 43 of which were required for UKCA coupling. These 652 additional diagnostics are the "unnecessary" ones, as it is possible to run the model without any diagnostic requests at all (except for those required for UKCA coupling). All fields that are required for UKCA coupling can be found in Tables 39 and 40 of Appendix C in UMDP 84 (Abraham *et al*., 2018).

The final GA7+StratTrop configuration includes 19 diagnostics from UKCA chemistry and aerosols that are used in the rose-stem tests, listed below. We did not include this information in the manuscript as we believed that these specifics were unnecessary as users can add or remove diagnostics from their own simulations as required.

| Diagnostic |
| --- |
| GLOMAP-mode accumulation mode (soluble) optical depth |
| Ozone ($O_3$) mass-mixing ratio (kg kg$^{-1}$) |
| GLOMAP-mode Aitken mode (soluble) organic matter mass-mixing ratio (kg kg$^{-1}$) |
| Age of air (s) |
| Cloud droplet number concentration (m$^{-3}$) |
| Nitrogen dioxide ($NO_2$) mass-mixing ratio from UKCA (kg kg$^{-1}$) |
| Primary emissions flux of sulphuric acid aerosol mass to GLOMAP-mode Aitken mode (soluble) (mol s$^{-1}$) |
| Geometric (number) mean dry diameter of particles in GLOMAP-mode accumulation mode (soluble) (m) |
| Vertically integrated plume scavenging of Cl in the GLOMAP-mode coarse mode (soluble) (mol s$^{-1}$) |
| Flux through $NO + RO_2$ reactions (mol s$^{-1}$) |
| Ozone dry deposition flux (mol s$^{-1}$) |
| $NO_y$ wet deposition flux (mol s$^{-1}$) |
| $NO_x$ emissions from lighting (mol s$^{-1}$) |
| NO surface emissions (kg m$^{-2}$ s$^{-1}$) |
| NO aircraft emissions (kg m$^{-2}$ s$^{-1}$) |
| Photolysis rate JO1D (s$^{-1}$) |
| Ozone ($O_3$) mass-mixing ratio on pressure levels (kg kg$^{-1}$) |
| Heaviside function on pressure levels |
| Flux through $NO + RO2$ reactions on pressure levels (mol s$^{-1}$) |

We have amended the text of the manuscript to read:

*The memory requirements can be lowered even more by turning off diagnostics and diagnostic sections that are not required*
5 *for UKCA coupling to reduce the number of large three-dimensions arrays that need to be allocated, leaving a model*
*configuration that only needs 4GB of memory.*

*Page 4, line 9: A short discussion of the differences in the ozone simulated by the two*
*configurations would be appropriate here. To put the changes into context, perhaps*
10 *these differences could be compared with the multi-model spread in any of the recent*
*model intercomparison exercises.*

We do not believe that such a quantitative or qualitative comparison would be useful in the context of the purpose of this
configuration and could in fact be misleading by implying that the fidelity of the output from the N48L38 configuration is

suitable for scientific studies. The changes to the model dynamics brought about by lowering of model resolution and model top mean that this configuration will rapidly deviate from a higher-resolution control. This low-resolution configuration is only able to be run for a few model hours, rather than the longer times that would be required for a meaningful study into model resolution, such as Stock *et al.* (2014). These results will also be highly biased by the initial conditions used.

*Table 1: I found the "Configuration details" of the black and purple curves slightly confusing.*
*Instead of giving the number of timesteps in one model hour, it seems more*
*natural to give the length of the timestep (eg. 20 minutes and 30 minutes).*

10    We have amended the table to:

*Table 1: Description of the curves presented in Figure 1.*

| Curve in Figure 1 | UM resolution | Configuration details | Peak memory usage (GiB) 1×1 MPI processes without OpenMP |
|---|---|---|---|
| *Green curve* | N48L70

3.75°×2.5° with 70 vertical levels up to 80km | GA6 configuration with 27 dynamical timesteps (20-minute dynamical timestep). | 1.62 |
| **Black curve** | N48L70

3.75°×2.5° with 70 vertical levels up to 80km | GA7+StratTrop configuration with 3 dynamical timesteps and 1 chemical timestep (20-minute dynamical timestep, 1-hour chemical timestep). | 12.13 |
| *Purple curve* | N48L38

3.75°×2.5° with 38 vertical levels up to 40km | GA7+StratTrop configuration with 2 dynamical timesteps and 1 chemical timestep (30-minute dynamical timestep, 1-hour chemical timestep). | 7.43 |
| *Blue curve* | N48L38

3.75°×2.5° with 38 vertical levels up to 40km | GA7+StratTrop configuration with 2 dynamical timesteps and 1 chemical timestep (30-minute dynamical timestep, 1-hour chemical timestep).
Only minimal diagnostic output is included. | 6.84 |
| *Red curve* | N48L38

3.75°×2.5° with 38 vertical levels up to 40km | GA7+StratTrop configuration with 2 dynamical timesteps and 1 chemical timestep (30-minute dynamical timestep, 1-hour chemical timestep).
Only minimal diagnostic output is included and the CFMIP Observation Simulator Package (COSP) has been disabled. | 3.99 |

*Figure 2: By plotting the results of the full and reduced configurations next to each other, the reader is required to eyeball the two figures to get a sense of the differences between the simulations. For panels (b) and (d), I would prefer to see the plot of the full configuration as-is, and then a plot of the difference between the reduced and full*

*versions. Also, what does "3-hour mean" mean in this context? Is this a single 3-hour*

*period from one particular day? More information is needed here.*

As mentioned, the changes in model dynamics mean that a quantitative difference between these two configurations is likely
to be misleading. Due to computational limitations, the low-resolution N48L38 model is only designed to be run for 3 model
hours, with all model output being given as 3-hour means. For comparison, the N96L85 configuration was sampled in the
same way so that a 3-hour mean could be plotted for both.

These configurations should never be compared in a quantitative way – the low resolution configuration on the VM is designed
only for model development, testing, and training, and should not be used for meaningful scientific studies. To emphasize the
differences in resolution Figures 2b and 2d have been replaced with surface plots over Europe, and the caption has been
updated. We have added the following text at the end of Section 4.1:

*Here both models were run for 3 model hours, with output provided as a mean over this period.*

Figure 2 is now:

[Figure]

*Figure 2: Comparison of modelled ozone from N96L85 and N48L38 UKCA configurations, highlighting the differences in model resolution
and vertical extent. All plots are 3-hour means from the first 3 model hours of simulation. (a, b) N96L85 UM-UKCA configuration. (c, d)
N48L38 UM-UKCA configuration.*

***Page 6, line 10: The first "to" should be "do".***

The text now reads:

5     *When developing a change often what is most important is the speed of compilation and the over-all turn-around from "what changes do I need to implement?" to "the model runs successfully".*

***Page 11, line 22: What is "mule-cumf"?***

10     As discussed in the reply to Reviewer #1, the "mule-cumf" tool allows for a bit-wise comparison of all fields contained in UM output files to be performed. However, in the context of this discussion these details are unnecessary, and so the reference to mule-cumf has been removed. The text now reads:

    *The output from these two jobs is then compared in the same way as for the KGO comparison.*

15

***Page 11, line 24: "of is desirable". I think you left out some words here.***

    Indeed, the text "MPI processes" is missing. The text now reads:

20     *Scientifically, maintaining results when changing the number (or decomposition) of MPI processes is desirable as it allows for the resources to be changed during long climate runs, allowing for more or less processes to be used without affecting the model output.*

***Page 13, line 1: Can you provide some kind of link to the online tutorials? Ideally in a***
25     ***permanently-available archive? Or could you zip them and add them as a supplement***
    ***to the paper?***

    All the references for the UKCA tutorials have DOIs that point to archived PDF files of the tutorial webpages that are held online in the University of Cambridge Apollo Repository (Abraham, 2013; Abraham et al., 2017; Abraham and Mann, 2014,
30     2016). The citation style for these references has been updated to include the live URL as well as the DOI, and these now read as:

    *Abraham, N. L.: UKCA & UMUI Tutorials for UM8.2, Online Learning Materials, doi:10.17863/CAM.22149 [online] Available from: http://www.ukca.ac.uk/wiki/index.php/UKCA_&_UMUI_Tutorials (Accessed 26 March 2018), 2013.*

*Abraham, N. L. and Mann, G. W.: UKCA Chemistry and Aerosol Tutorials for UM8.4, Online Learning Materials, doi:10.17863/CAM.22151 [online] Available from: http://www.ukca.ac.uk/wiki/index.php/UKCA_Chemistry_and_Aerosol_Tutorials (Accessed 26 March 2018), 2014.*

*Abraham, N. L. and Mann, G. W.: UKCA Chemistry and Aerosol Tutorials at vn10.4 using Rose & Cylc, Online Learning*
5 *Materials, doi:10.17863/CAM.22152 [online] Available from: http://www.ukca.ac.uk/wiki/index.php/UKCA_Chemistry_and_Aerosol_Tutorials_at_vn10.4 (Accessed 26 March 2018), 2016.*

*Abraham, N. L., Bellouin, N. and Schmidt, A.: UKCA Chemistry and Aerosol Tutorials at vn10.9 using Rose & Cylc, Online Learning Materials, doi:10.17863/CAM.22153 [online] Available from:*
10 *http://www.ukca.ac.uk/wiki/index.php/UKCA_Chemistry_and_Aerosol_Tutorials_at_vn10.9 (Accessed 26 March 2018), 2017.*

**Additional Changes**

In addition to the above changes, the following changes have also been made:

15 • The reference for RCMES has been updated from NASA Jet Propulsion Laboratory (2018) to Lee et al. (2018):

*Lee, H., Goodman, A., McGibbney, L., Waliser, D., Kim, J., Loikith, P., Gibson, P. and Massoud, E.: Regional Climate Model Evaluation System powered by Apache Open Climate Workbench v1.3.0: an enabling tool for facilitating regional climate studies, Geosci. Model Dev. Discuss., 1–23, doi:10.5194/gmd-2018-113, 2018.*

20
• The in-text citation for the version 10.9 tutorials has been changed slightly within the text of the Code Availability section, although the reference has remained the same

*Abraham et al. (2017)*

25
• The following has been added to the acknowledgments:

*The work of MD was supported by the Met Office Hadley Centre Climate Programme funded by BEIS and Defra.*

**References**

30 Abraham, N. L.: UKCA & UMUI Tutorials for UM8.2, Online Learning Materials, doi:10.17863/CAM.22149 [online]

Available from: http://www.ukca.ac.uk/wiki/index.php/UKCA_&_UMUI_Tutorials (Accessed 26 March 2018), 2013.

Abraham, N. L. and Mann, G. W.: UKCA Chemistry and Aerosol Tutorials for UM8.4, Online Learning Materials, doi:10.17863/CAM.22151 [online] Available from: http://www.ukca.ac.uk/wiki/index.php/UKCA_Chemistry_and_Aerosol_Tutorials (Accessed 26 March 2018), 2014.

5    Abraham, N. L. and Mann, G. W.: UKCA Chemistry and Aerosol Tutorials at vn10.4 using Rose & Cylc, Online Learning Materials, doi:10.17863/CAM.22152 [online] Available from: http://www.ukca.ac.uk/wiki/index.php/UKCA_Chemistry_and_Aerosol_Tutorials_at_vn10.4 (Accessed 26 March 2018), 2016.

Abraham, N. L., Bellouin, N. and Schmidt, A.: UKCA Chemistry and Aerosol Tutorials at vn10.9 using Rose & Cylc, Online
10   Learning Materials, doi:10.17863/CAM.22153 [online] Available from: http://www.ukca.ac.uk/wiki/index.php/UKCA_Chemistry_and_Aerosol_Tutorials_at_vn10.9 (Accessed 26 March 2018), 2017.

Abraham, N. L., Archibald, A. T., Bellouin, N., Boucher, O., Braesicke, P., Bushell, A., Carslaw, K., Collins, B., Dalvi, M., Dennison, F., Emmerson, K., Folberth, G., Haywood, J., Hewitt, A., Johnson, C., Kipling, Z., Macintyre, H., Mann, G., Telford,
15   P., Merikanto, J., Morgenstern, O., O'Connor, F., Ordonez, C., Osprey, S., Pringle, K., Pyle, J., Rae, J., Reddington, C., Savage, N., Sellar, A., Spracklen, D., Stier, P., West, R., Mulcahy, J., Woodward, S., Boutle, I. and Woodhouse, M. T.: UMDP 084: United Kingdom Chemistry and Aerosol (UKCA) Technical Description. [online] Available from: https://code.metoffice.gov.uk/doc/um/vn11.0/papers/umdp_084.pdf (Accessed 23 February 2018), 2018.

Collins, W. J., Bellouin, N., Doutriaux-Boucher, M., Gedney, N., Halloran, P., Hinton, T., Hughes, J., Jones, C. D., Joshi, M.,
20   Liddicoat, S., Martin, G., O'Connor, F., Rae, J., Senior, C., Sitch, S., Totterdell, I., Wiltshire, A. and Woodward, S.: Development and evaluation of an Earth-System model – HadGEM2, Geosci. Model Dev., 4(4), 1051–1075, doi:10.5194/gmd-4-1051-2011, 2011.

Cresswell, P.: Unified Model Documentation Paper X10: Unified Model Virtual Machine Guide. [online] Available from: https://code.metoffice.gov.uk/doc/um/vn11.0/papers/umdp_X10.pdf (Accessed 6 March 2018), 2018.

25   Lee, H., Goodman, A., McGibbney, L., Waliser, D., Kim, J., Loikith, P., Gibson, P. and Massoud, E.: Regional Climate Model Evaluation System powered by Apache Open Climate Workbench v1.3.0: an enabling tool for facilitating regional climate studies, Geosci. Model Dev. Discuss., 1–23, doi:10.5194/gmd-2018-113, 2018.

Matthews, D. P.: Vagrant virtual machines with FCM + Rose + Cylc installed, [online] Available from: https://github.com/metomi/metomi-vms, 2018.

30   NASA Jet Propulsion Laboratory: RCMES: Regional Climate Model Evaluation System, [online] Available from: https://rcmes.jpl.nasa.gov/ (Accessed 27 March 2018), 2018.

O'Connor, F. M., Johnson, C. E., Morgenstern, O., Abraham, N. L., Braesicke, P., Dalvi, M., Folberth, G. A., Sanderson, M. G., Telford, P. J., Voulgarakis, A., Young, P. J., Zeng, G., Collins, W. J. and Pyle, J. A.: Evaluation of the new UKCA climate-composition model-Part 2: The troposphere, Geosci. Model Dev., 7(1), doi:10.5194/gmd-7-41-2014, 2014.

Stock, Z. S., Russo, M. R. and Pyle, J. A.: Representing ozone extremes in European megacities: the importance of resolution in a global chemistry climate model, Atmos. Chem. Phys., 14(8), 3899–3912, doi:10.5194/acp-14-3899-2014, 2014.

U.K. Met Office: Met Office Science Repository Service, [online] Available from: https://code.metoffice.gov.uk/trac/home/ (Accessed 10 April 2018), 2018.

---

## Author Comment (AC2) · 20 Jul 2018

Many thanks for your positive and helpful comments. Please note that a detailed response to your review can be found here:

https://www.geosci-model-dev-discuss.net/gmd-2018-125/gmd-2018-125-AC1-supplement.pdf